# Understanding contributors to racial and ethnic inequities in COVID-19 incidence and mortality rates

Karen E. Joynt Maddox[1,2,3]*, Mat Reidhead[3], Joshua Grotzinger[3], Timothy McBride[2,4], Aaloke Mody[5], Elna Nagasako[6], Will Ross[7], Joseph T. Steensma[4], Abigail R. Barker[2,4]

1 Cardiovascular Division, Washington University School of Medicine, St. Louis, Missouri, United States of America, 2 Center for Health Economics and Policy, Institute for Public Health at Washington University, St. Louis, Missouri, United States of America, 3 Washington University Brown School, St. Louis, Missouri, United States of America, 4 Missouri Hospital Association, Jefferson City, Missouri, United States of America, 5 Division of Infectious Diseases, Washington University School of Medicine, St. Louis, Missouri, United States of America, 6 Honolulu, Hawaii, United States of America, 7 Division of Nephrology, Washington University School of Medicine, St. Louis, Missouri, United States of America

* kjoyntmaddox@wustl.edu

## Abstract

### Background

Racial inequities in Coronavirus 2019 (COVID-19) have been reported over the course of the pandemic, with Black, Hispanic/Latinx, and Native American individuals suffering higher case rates and more fatalities than their White counterparts.

### Methods

We used a unique statewide dataset of confirmed COVID-19 cases across Missouri, linked with historical statewide hospital data. We examined differences by race and ethnicity in raw population-based case and mortality rates. We used patient-level regression analyses to calculate the odds of mortality based on race and ethnicity, controlling for comorbidities and other risk factors.

### Results

As of September 10, 2020 there were 73,635 confirmed COVID-19 cases in the State of Missouri. Among the 64,526 case records (87.7% of all cases) that merged with prior demographic and health care utilization data, 12,946 (20.1%) were Non-Hispanic (NH) Black, 44,550 (69.0%) were NH White, 3,822 (5.9%) were NH Other/Unknown race, and 3,208 (5.0%) were Hispanic. Raw cumulative case rates for NH Black individuals were 1,713 per 100,000 population, compared with 2,095 for NH Other/Unknown, 903 for NH White, and 1,218 for Hispanic. Cumulative COVID-19-related death rates for NH Black individuals were 58.3 per 100,000 population, compared with 38.9 for NH Other/Unknown, 19.4 for NH White, and 14.8 for Hispanic. In a model that included insurance source, history of a social determinant billing code in the patient's claims, census block travel change, population

**Data Availability Statement:** Data cannot be shared publicly because of HIPAA and data use agreements that prohibit the public sharing of protected health information. Data are available

from the Hospital Industry Data Institute (contact via www.hidionline.com/hidi/Resources.aspx) under appropriate data use agreements and human subjects review processes for researchers who wish to apply for access to confidential data.

**Funding:** The authors received no specific funding for this work.

**Competing interests:** Dr. Joynt Maddox receives research support from the National Heart, Lung, and Blood Institute (R01HL143421) and National Institute on Aging (R01AG060935, R01AG063759, and R21AG065526), and previously did contract work for the US Department of Health and Human Services. The other authors report no disclosures. This does not alter our adherence to PLOS ONE policies on sharing data and materials.

density, Area Deprivation Index, and clinical comorbidities, NH Black race (OR 1.75, 1.51–2.04, p<0.001) and NH Other/Unknown race (OR 1.83, 1.36–2.46, p<0.001) remained strongly associated with mortality.

## Conclusions

In Missouri, COVID-19 case rates and mortality rates were markedly higher among NH Black and NH Other/Unknown race than among NH White residents, even after accounting for social and clinical risk, population density, and travel patterns during COVID-19.

## Introduction

Racial inequities in case rates and clinical outcomes from the Coronavirus 2019 (COVID-19) have been reported over the course of the pandemic, with Black, Hispanic/Latinx, and Native American individuals suffering markedly higher case rates and more fatalities than their White counterparts [1]. There are a number of potential contributors to racial inequities in COVID-19 [2]. These include risk factors for exposure to the disease such as essential worker status, use of public transportation, and population density [3–5], as well as risk factors for poor outcomes among people with established disease, including comorbidities and social determinants of health such as poverty, education, access to care, and insurance coverage that are known to be associated with adverse health outcomes more broadly [6–11].

Understanding diverse contributors to these inequities and their mechanisms is a crucial step in designing clinical and policy initiatives that might alleviate them. Specifically, if the majority of the excess risk associated with minority race or ethnicity is a result of differential exposure to the disease, and therefore differential case rates, interventions might include better protection for essential workers in terms of personal protective equipment or paid sick leave when quarantining, or better access to testing among certain groups. On the other hand, if the majority of the excess risk is a result of differential clinical outcomes among those who contract the disease, better attention to access to novel therapies, access to high-quality health care, or a broader focus on improving health among at-risk groups would be more promising strategies.

Prior studies examining racial and ethnic inequities in COVID-19 case rates and clinical outcomes have often been limited to inpatients, or to either case rates or clinical outcomes but not both. Using a unique statewide dataset of confirmed COVID-19 cases across Missouri, linked with historical statewide hospital data, we examined differences by race and ethnicity in raw population-based case and mortality rates, and then used patient-level regression analyses to calculate the odds of mortality based on race and ethnicity. We also aimed to understand which risk factors, specifically social and clinical risk, population density, and travel patterns during COVID-19, were most strongly associated with racial and ethnic differences in COVID-19 incidence and outcomes.

## Methods

### Data

Case and fatality records were obtained from the Missouri Department of Health and Senior Services (DHSS) from March 1 to September 20, 2020. Cases were linked to administrative hospital claims data from the Hospital Industry Data Institute, a statewide repository of all

inpatient and emergency department (ED) encounters in the state. We used a master patient identifier from Novetta Entity Analytics (Novetta LLC, McLean, VA) based on patient name, address, date of birth and gender to match individuals with confirmed COVID-19 to one or more prior hospital encounters during the previous two-year period. We successfully matched 64,526 out of 73,635 cases, for a match rate of 87.7% (see **S1 Fig** for flow diagram). This linkage allowed us to determine demographics (age, race, ethnicity), payer information, and home ZIP code among any person with any prior health care encounter in an inpatient or outpatient hospital setting, regardless of whether he or she were hospitalized for COVID-19. From these clinical data, we also extracted information on comorbidities as defined using the Chronic Conditions Warehouse methodology from the U.S. Centers for Medicare and Medicaid Services, and whether or not the patient had ever had a billing code indicative of a social determinant of health (SDOH) challenge using International Classification of Diseases, Tenth Revision, Clinical Modification (ICD-10-CM) z-codes Z55-Z65, which include codes for housing instability, lack of social support, history of abuse, and other important issues [12, 13].

We used patient address to geocode confirmed COVID-19 cases to the census block group-level and ZIP Code Tabulation Area-level to merge the case and hospital encounter data with 2015 Area Deprivation Index data from the University of Wisconsin Neighborhood Atlas [14], and 2018 population and population density estimates from the U.S. Census American Community Survey [15]. We also linked the data with 2019–2020 mobility data from the SafeGraph COVID-19 Data Consortium [16], Social Distancing Metrics, which links cellular phone locations to points of interest and allowed us to determine whether, in each census block group, the median time spent at home went up or down during the first four months of the pandemic in Missouri (March-June, 2020) compared to the same time period during 2019, a proxy for whether people were able to restrict travel.

### Predictors and outcomes

Our primary predictor was race/ethnicity as obtained from individuals' hospital data, defined as Non-Hispanic Black or African-American (henceforth NH Black), Non-Hispanic White (henceforth NH White), Hispanic, and Non-Hispanic Other or Unknown race (henceforth NH Other/Unknown) which were the categories available from the harmonized hospital dataset, and reflect hospital registration data. The Other or Unknown group includes six individual race categories (American Indian/Alaska Native, Asian, Pacific Islander, Other, Multiple Races, and Unknown or Refused), in addition to individuals with multiple races reported in the hospital claims data where no dominant mode was detected (e.g. an individual with two claims, one coded White and one coded Asian, **S1 Table**). We considered census blocks in the highest two quintiles of Area Deprivation Index to be highly deprived, and those in the highest quintile of population density to be highly dense. Our primary outcomes were case rate per 100,000 population and case fatality rates using deceased vital status codes included in the DHSS data, which include COVID-19-related expirations occurring in any setting.

### Statistical analysis

We first calculated patient characteristics overall and by race/ethnicity group. We then plotted cumulative case and mortality rates by race/ethnicity over the study period. We then estimated multivariable associations of demographic, community, SDOH and clinical covariates on the log-odds of experiencing COVID-19 related mortality using logistic and hierarchical generalized linear methods. Three models were fit under each specification to evaluate stepwise differences in fixed and random effects for race and ethnicity associated with each model: demographic only; demographic and SDOH; and demographic, SDOH and clinical factors.

Our primary analysis included race and ethnicity as fixed effects in our logistic models to quantify the log-odds of case fatality associated with NH Black, NH Other/Unknown and Hispanic individuals compared to White cases.

In secondary analyses, in order to estimate the extent to which observed disparities in COVID-19 related mortality persisted after accounting for individual risk factors, we constructed a ratio that captures the excess risk of belonging to a non-White racial group (beyond the risks that relate to covarying individual characteristics). To do this, we estimated mixed models in which race and ethnicity are modeled as random effects, to capture the individual-level risks within each racial group (the rate predicted by the model, which allows for explanatory variables that often covary by race). We also calculated the expected rate holding individual risk constant (i.e. the expected value determined by evaluating the linear combination of the individual's other demographic and comorbidity data and the coefficient vector). When aggregated over all members of a racial group, this yields the expected rate based only on non-racial explanatory variables. The ratio of predicted-to-expected case fatality rates for each race/ethnicity category provides an estimate of what may be termed the "otherwise unexplained" racial disparities in COVID-19-related mortality.

This study was approved by the Office of Human Research Protection at the Washington University School of Medicine. The requirement for informed consent and HIPAA notification were waived due to the de-identified nature of the data. SAS version 9.4, Cary North Carolina was used for all analyses. For each primary outcome, a p value less than 0.05/3 = 0.017 was considered statistically significant to account for multiple comparisons.

## Results

### Case characteristics

As of September 10, 2020 there were 73,600 confirmed COVID-19 cases in the State of Missouri. Among the 64,526 case records (87.7% of all cases) that merged with prior demographic and health care utilization data, 12,946 (20.1%) were NH Black, 44,550 (69.0%) were NH White, 3,822 (5.9%) were NH Other/Unknown, and 3,208 (5.0%) were Hispanic (**Table 1**). NH White patients were the oldest on average (47.2 years, compared to 44.4 for NH Black, 38.2 for NH Other/Unknown, and 37.2 for Hispanic patients), and roughly 40% of each group was male. NH White patients were also the least likely to be insured by Medicaid (8.0%, compared to 23.2% for NH Black patients, 19.0% for NH Other/Unknown patients, and 28.3% for Hispanic patients) or to be uninsured (12.7%, compared to 29.4% for NH Black patients, 24.1% for NH Other/Unknown patients, and 33.1% for Hispanic patients). NH White and NH Other/Unknown patients were less likely to have SDOH Z-codes in their billing records, indicating a code for a relevant social risk factor, at 3.8% and 4.0% respectively, compared to 9.5% for NH Black patients and 5.6% for Hispanic patients. The 9,074 cases that did not merge with prior hospital records were younger, more often male, and particularly likely to be of other or unknown race (reflecting data collected with the test itself rather than via our demographic matching).

Regarding comorbidities, NH White patients were the most likely to have evidence of a psychiatric disorder, but less likely to have most other comorbidities including asthma, chronic kidney disease, heart failure, hypertension, obesity, tobacco use, or uncontrolled diabetes than NH Black patients. Patterns for other-race patients and Hispanic patients were variable.

For neighborhood characteristics, NH Black and Hispanic patients were much more likely to live in areas with high levels of deprivation as defined by the Area Deprivation Index, and to live in high-population-density areas. White patients were the most likely to live in census block groups where the median time spent at home, as measured by cell phone activity, was

**Table 1. Patient characteristics by race and ethnicity.**

| | Non-Hispanic | | | Hispanic | Total | Unmerged with prior hospital records |
| --- | --- | --- | --- | --- | --- | --- |
| | **Black or African American** | **White** | **Other / Unknown Races** | | | |
| COVID-19 Cases (n) | 12,946 | 44,550 | 3,822 | 3,208 | 64,526 | 9,074 |
| Number of deaths | 425 | 911 | 62 | 39 | 1437 | 717 |
| Case-Fatality Rate | 3.3%* | 2.0%* | 1.6%^ | 1.2%* | 2.2% | 0.79%* |
| DEMOGRAPHICS | | | | | | |
| Age (mean, in years) | 44.4* | 47.2* | 38.2* | 37.2* | 45.6 | 37.0* |
| Male | 37.2%* | 39.1% | 41.2%^ | 41.9%^ | 39.0% | 52.0%* |
| Female | 62.8%* | 60.9% | 58.8%^ | 58.1%^ | 61.0% | 48.0%* |
| NH Black | 100.0% | 0.0% | 0.0% | 0.0% | 20.1% | 11.1%* |
| NH White | 0.0% | 100.0% | 0.0% | 0.0% | 69.0% | 45.0%* |
| NH Other/Unknown | 0.0% | 0.0% | 100.0% | 0.0% | 5.9% | 38.6%* |
| Hispanic | 0.0% | 0.0% | 0.0% | 100.0% | 5.0% | 5.3% |
| INDIVIDUAL SOCIAL DETERMINANTS | | | | | | |
| Medicaid | 23.2%* | 8.0%* | 19.0%* | 28.3%* | 12.7% | na |
| Uninsured | 29.4%* | 12.7%* | 24.1%* | 33.1%* | 17.7% | na |
| SDOH Z-Code | 9.5%* | 3.8%* | 4.0%^ | 5.6% | 5.0% | na |
| INDIVIDUAL COMORBIDITIES | | | | | | |
| Psychiatric Disorder | 20.9%^ | 23.2%* | 12.3%* | 18.0%* | 21.9% | na |
| Alzheimer's/Dementia | 5.9% | 5.9%* | 2.5%* | 3.1%* | 5.6% | na |
| Asthma | 14.4%* | 6.9%* | 5.4%* | 7.1%^ | 8.3% | na |
| Chronic Kidney Disease | 20.6%* | 13.0%* | 8.7%* | 11.8%* | 14.2% | na |
| COPD | 8.1% | 8.2%* | 3.3%* | 5.3%* | 7.8% | na |
| Heart Failure | 8.5%* | 6.0%^ | 2.6%* | 3.5%* | 6.2% | na |
| Hypertension | 41.6%* | 31.0%* | 18.3%* | 21.6%* | 31.9% | na |
| Lung Cancer | 0.4% | 0.5%^ | 0.1%^ | 0.3% | 0.4% | na |
| Obesity | 22.3%* | 12.6%* | 8.5%* | 12.0%* | 14.3% | na |
| Tobacco Use | 16.2%* | 11.1%* | 8.7%* | 9.2%* | 11.9% | na |
| Uncontrolled Diabetes | 9.2%* | 4.6%* | 3.7%* | 6.1% | 5.6% | na |
| AREA-LEVEL DATA | | | | | | |
| Census Block Group Area Deprivation Index q4-5 | 69.4%* | 44.2%* | 56.5%* | 70.7%* | 51.3% | 47.0%* |
| Census Block Group More Time Spent at Home (Mar-Jun 2020 vs. 2019) | 67.4%* | 75.9%* | 71.9%^ | 65.7%* | 73.5% | 75.8%* |
| Urban (highest quintile of ZIP Code Population Density) | 90.8%* | 56.7%* | 70.3%* | 67.8%^ | 64.9% | 67.0%* |

Source: Authors' analysis of COVID-19 case data from the Missouri Department of Health and Senior Services merged with 2018–2020 hospital claims data from the Hospital Industry Data Institute, 2015 Area Deprivation Index data from the University of Wisconsin Neighborhood Atlas, 2019–2020 mobility data from Safe Graph, and 2018 population estimates from the U.S. Census American Community Survey. Abbreviations: SDOH, Social Determinant of Health; q, Quintile; COPD, Chronic Obstructive Pulmonary Disease.

*P<0.001

^P<0.05.

higher in March-June 2020 than March-June 2019, suggesting higher levels of change in travel patterns associated with local stay-at-home orders.

## Population-based case and mortality rates

Examining case rates over time, there were early and persistent inequities across racial and ethnic groups, particularly for the NH Black and NH Other/Unknown groups (**Fig 1**).

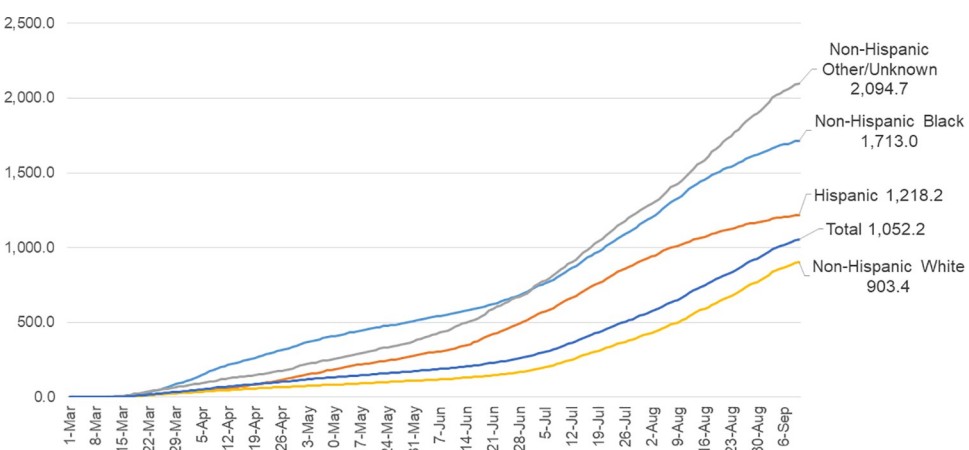

**Fig 1. Cumulative COVID-19 related cases per 100,000 by race and ethnicity, Missouri residents, Mar. 1, 2020 – Sep. 10, 2020.** Source: Authors' analysis of COVID-19 case data from the Missouri Department of Health and Senior Services and 2018 population estimates from the U.S. Census American Community Survey.

Patterns for population-based mortality were similar (**Fig 2**). Cumulatively, the NH Black COVID-19-related death rate was 58.3 per 100,000 population in the state, compared with 38.9 for NH Other/Unknown, 19.4 for NH White, and 14.8 for Hispanic. These inequities appeared within the first month of the pandemic and grew in magnitude until mid-May 2020, with the gap between Black and White persisting thereafter.

## Multivariate regression models for mortality

There were 1,437 deaths in the sample, for a raw mortality rate of 2.2%. In logistic regression models that included demographics alone, age (odds ratio [OR] 1.09, 95% confidence interval

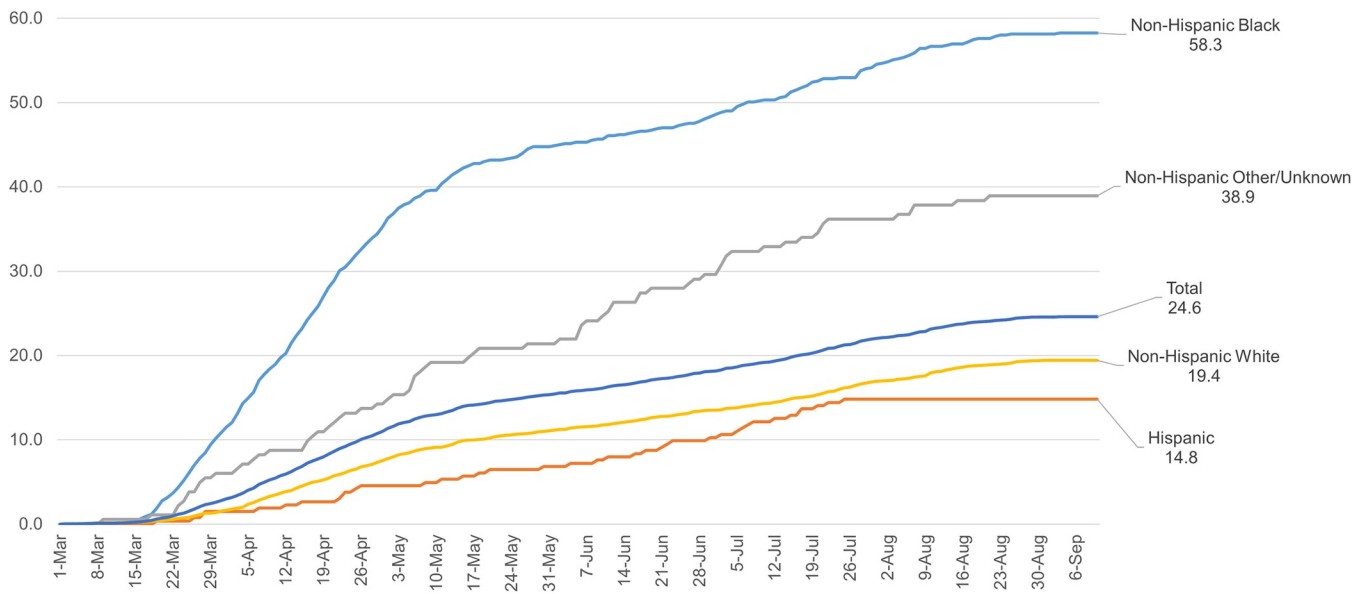

**Fig 2. Cumulative COVID-19 related deaths per 100,000 by race and ethnicity, Missouri residents, Mar. 1, 2020 –Sep. 10, 2020.** Source: Authors' analysis of COVID-19 case data from the Missouri Department of Health and Senior Services and 2018 population estimates from the U.S. Census American Community Survey.

**Table 2. Logistic regression models for mortality.**

| | Demographic Model | Demographic and Neighborhood Model | Demographic, Neighborhood, and Comorbidity Model |
|---|---|---|---|
| **Effect** | OR (95% CI) | OR (95% CI) | OR (95% CI) |
| **Age (continuous)** | 1.09 (1.09–1.10) | 1.09 (1.09–1.10) | 1.07 (1.07–1.08) |
| **Male** | 2.00 (1.78–2.24) | 2.00 (1.78–2.24) | 1.93 (1.71–2.17) |
| **Non-Hispanic Black** | 2.82 (2.48–3.21) | 2.07 (1.79–2.39) | 1.75 (1.51–2.04) |
| **Non-Hispanic Other/Unknown** | 1.78 (1.34–2.36) | 1.58 (1.18–2.10) | 1.83 (1.36–2.46) |
| **Hispanic** | 1.54 (1.09–2.16) | 1.32 (0.93–1.86) | 1.25 (0.88–1.78) |
| **Medicaid** | | 1.29 (1.01–1.64) | 1.06 (0.83–1.35) |
| **Uninsured** | | 0.90 (0.72–1.13) | 0.98 (0.78–1.23) |
| **SDOH Z-Code** | | 2.02 (1.69–2.41) | 1.01 (0.84–1.22) |
| **Census Block Group Area Deprivation Index q4-5** | | 1.07 (0.95–1.22) | 1.05 (0.92–1.19) |
| **Census Block Group Less Median Time Spent at Home (Mar-Jun 2020 vs. 2019)** | | 0.86 (0.74–0.99) | 0.85 (0.73–0.99) |
| **ZIP Code Population Density q5** | | 2.14 (1.83–2.49) | 1.79 (1.53–2.10) |
| **Psychiatric Disorder** | | | 1.28 (1.13–1.46) |
| **Alzheimer's/Dementia** | | | 2.23 (1.95–2.55) |
| **Asthma** | | | 0.88 (0.71–1.10) |
| **Chronic Kidney Disease** | | | 2.75 (2.38–3.19) |
| **COPD** | | | 1.23 (1.06–1.42) |
| **Heart Failure** | | | 1.27 (1.11–1.46) |
| **Hypertension** | | | 1.10 (0.92–1.31) |
| **Lung Cancer** | | | 2.39 (1.64–3.47) |
| **Obesity** | | | 1.30 (1.12–1.50) |
| **Tobacco Use** | | | 1.19 (1.00–1.42) |
| **Uncontrolled Diabetes** | | | 1.27 (1.09–1.49) |
| **Observations** | 64,526 | 64,526 | 64,526 |
| **COVID-19-Related Deaths (dep. variable)** | 1,437 | 1,437 | 1,437 |
| **C-Statistic** | 0.912 | 0.918 | 0.936 |

Source: Authors' analysis of COVID-19 case data from the Missouri Department of Health and Senior Services merged with 2018–2020 hospital claims data from the Hospital Industry Data Institute, 2015 Area Deprivation Index data from the University of Wisconsin Neighborhood Atlas, 2019–2020 mobility data from Safe Graph, and 2018 population estimates from the U.S. Census American Community Survey. Abbreviations: SDOH, Social Determinant of Health; q, Quintile; COPD, Chronic Obstructive Pulmonary Disease.

[CI] 1.09–1.10 per one year increase in age) and male sex (OR 2.00 (1.78–2.24)) were associated with higher odds of mortality (**Table 2**). NH Black race was associated with a 2.82 times higher odds of dying (2.48–3.21), NH Other/Unknown race with 1.78 times higher odds (1.34–2.36), and Hispanic ethnicity with 1.54 times higher odds (1.09–2.16) of dying as compared to NH White individuals. After adding risk factors including insurance source, history of an SDOH Z-code in the patient's claims, census block travel change, population density, and Area Deprivation Index to the model, NH Black race (OR 2.07, 1.79–2.39) and NH Other/Unknown race (OR 1.58, 1.18–2.10) remained associated with higher odds of mortality. Finally, after adding clinical comorbidities to the model, NH Black race (OR 1.75, 1.51–2.04) and NH Other/Unknown race (OR 1.83, 1.36–2.46) remained strongly associated with higher odds of mortality.

The ratio analyses based on random effects models revealed similar results. Observed case fatality rates for NH Black cases were 1.6 times the rate of NH White cases (**Fig 3**). After accounting for age and sex, risk-adjusted case fatality rates for NH Black cases increased to 2.4

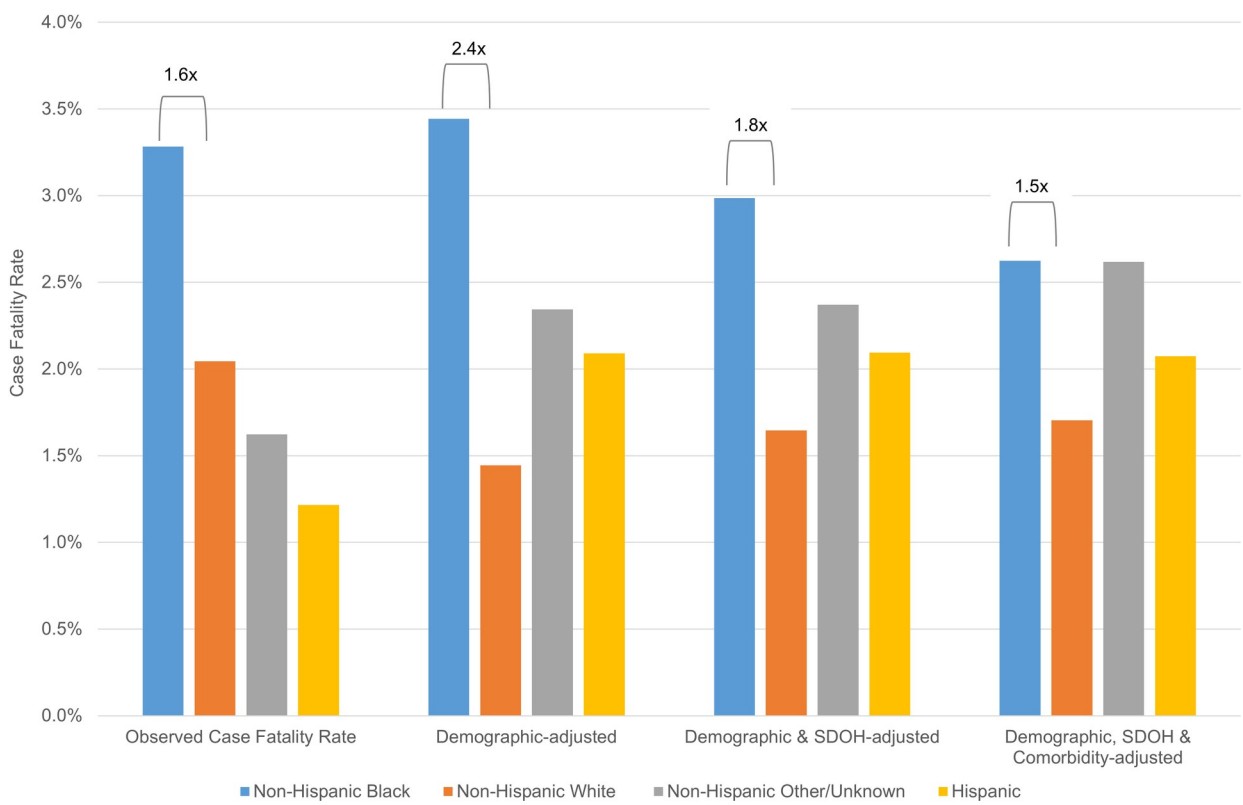

**Fig 3. Observed and risk-adjusted case fatality rates.** Abbreviations: SDOH, social determinants of health. Brackets depict ratio of case fatality rate for the group of interest versus White patients, using the random effects model described in the methods.

times the rate for NH White cases. Case fatality disparities were moderated but persisted after accounting for demographic and social factors (NH Black case fatality 1.8 times NH White case fatality) and more so when clinical characteristics were added to the model (NH Black case fatality 1.5 times NH White case fatality). Analyses of the intraclass coefficient from each model suggested that race explained roughly 5.4% or the variability in mortality in a model that included demographics alone, but that figure dropped to 2.9% when SDOH were added to the model, and 2.4% when clinical comorbidities were added (**S2 Table**).

## Discussion

We found that major inequities in case rates and mortality for confirmed COVID-19 cases across Missouri emerged early and persisted well into the pandemic. Confirmed case rates for NH Black Missourians were markedly higher than for their NH White counterparts, as were case and mortality rates among the relatively small number of NH Other/Unknown race patients in our dataset. Additionally, analysis of patient-level data suggested that NH Black and NH Other/Unknown race were strongly associated with a higher risk of mortality, even after accounting for age and sex, social risk, neighborhood characteristics, mobility as assessed by cell phone data, and medical comorbidities.

The reasons for the markedly different case diagnosis rate by race and ethnicity are likely many. Black and Hispanic individuals, as well as those of other racial groups, are more likely to be essential workers, and to be in high-contact essential employment positions, like health care, education, and public transportation, as well as to use public transportation for job access

[3, 17]. NH Black and Hispanic individuals also live in more population-dense ZIP codes and denser housing conditions, on average, than NH White individuals in Missouri, which can also impact one's likelihood of coming into contact with someone with known or unknown COVID-19 infection. Another possible mediator of exposure that we were not able to measure in our study is the prevalence of intergenerational households [18]. The nature and number of intergenerational contacts varies by culture, which has implications for exposure to COVID-19 [19]. It is possible that the high case rates seen in the NH Black and NH Other/Unknown race populations in Missouri were related to exposure through one or more of these means, though the lower case rates among Hispanic individuals counters this hypothesis somewhat.

In patient-level regression analyses, mortality was much higher among Black and Other race individuals than White ones, with odds ratios near two for mortality among confirmed cases even after accounting for a host of potential confounders and mediators, including social risk factors like insurance status and Area Deprivation Index as well as clinical comorbidities. It is likely that the residual differences we see by race even after including explanatory variables reflect not biological differences in likelihood of poor outcomes based on skin color per se, but differences in likelihood of poor outcomes because of the many downstream impacts of exposure to individual and structural racism, though we cannot test each of these directly. For example, there is convincing evidence that the experience of racism and discrimination leads to inflammation and elevated levels of circulating biomarkers associated with poor health outcomes, which may hinder people's ability to fight COVID-19 [20–23]. Racism and discrimination are also associated with poor health in general; claims-based comorbidity assessment cannot determine differences in risk within diagnoses, for example failing to differentiate between severe longstanding diabetes and more mild or recent diabetes, risk differentials that might inaccurately capture clinical severity particularly in minority groups.

It is also possible that worse outcomes for NH Black and NH Other/Unknown race individuals represent interpersonal racism leading to poorer quality care within the same hospitals, for example, clinicians electing to treat Black or other race patients differently from White patients. While treatment data were unavailable in our dataset, future research is needed to examine the types of treatments that Black and Other race patients compared to White patients have received over the course of the pandemic, and to ensure there has been equity in access to clinical trials and other treatments such as convalescent plasma.

The choice of hospital may matter as well: one recent study suggested that much of the difference in mortality between Black and White patients might be explained by the hospitals in which each group receives care [24]. Prior studies have shown that Black and Hispanic individuals tend to receive care in lower-quality hospitals, which could reflect structural racism in differential resourcing of hospitals that traditionally serve these groups, or other barriers to accessing high-quality care such as transportation [25]. Transportation and access issues may be particularly acute in rural parts of Missouri and other states; rural areas of the state are disproportionately White, suggesting that disparities might be even greater in urban areas where access to referral centers is higher.

Another possibility for the patterns we report is that cases may be differentially confirmed by testing based on race or ethnicity. While we had a statewide sample of cases regardless of hospitalization status, it is likely that many cases of COVID-19 go un-diagnosed. Prior studies have shown that access to testing differs by race, which could mean that more cases go unconfirmed in communities of color. If diagnosis rates differ by race, this could create selection bias, whereby the Black, Hispanic, or Other/Unknown race patients in our sample are comprised of a sicker, more symptomatic subset of total COVID-19 cases. Worse outcomes among confirmed cases in these groups would in that case represent selection rather than biology. However, our population-based analyses, which demonstrate markedly higher fatality rates

per 100,000 population, show similar if not more striking disparities, and these numbers should not suffer from selection bias to the same degree.

Our study should be interpreted in light of other extant findings in the literature. A study of over 11,000 patients from the Premier network found no significant differences in outcomes for hospitalized patients by race after adjusting for comorbidities, neighborhood characteristics, and insurance status [26]. Studies from the Veterans Health Administration reported a higher incidence of testing and hospitalization among Black veterans but no difference in mortality by race after adjusting for clinical characteristics [27, 28]. Studies from New York City and Wisconsin found no difference in mortality among Black patients after accounting for comorbidities and presenting severity of illness [29]. We suspect that the differences in these findings compared to ours reflect the use of clinical data to adjust risk and our use of a statewide rather than hospital-based sample, though we cannot be sure.

Our study has limitations. We used hospital registration data to obtain race and ethnicity data, which have a number of limitations. Because we relied on claims data, we lack detailed clinical information such as blood pressure or oxygen saturation at admission with which to risk adjust. Area Deprivation Index and American Community Survey data are only updated periodically, and shifts in demographics may have occurred between 2015 and the present. Some individuals lack cell phones, which could bias ZIP code-level cell phone data. Our data reflect the experience of a single state; while it is a diverse state with urban and rural areas, a sizeable minority population, and a range of hospital types, patterns we noted here may not apply to other states with different demographics, a different time course of COVID-19, or different patterns or mitigators of structural racism compared to Missouri. Our NH Other/Unknown and Hispanic groups were small, and likely reflect a wide diversity of nationalities, backgrounds, and socioeconomic circumstances. Our findings regarding Hispanic individuals run counter to findings in other parts of the country with larger Hispanic populations, and therefore should be interpreted with caution. Data limitations and sample size precluded further breakdown of these groups into subgroups by country of origin or immigration status, but such detailed analysis is an important direction for future work. We were unable to match roughly 12 percent of cases to prior health care utilization in hospital inpatient or outpatient departments, and this group undoubtedly represents a healthier subset of cases, with lower raw mortality rates. If members of racial or ethnic minority groups were less likely to have health care interactions prior to their COVID-19 diagnosis due to limitations in access, whether due to financial or insurance-related concerns, this could bias our sample, as well as limit our full ascertainment of comorbidities. Finally, death data may lag in reporting.

Our findings suggest that, to reduce or eliminate such inequities in future waves of COVID-19 or in future pandemics, attention needs to be paid to reducing exposure among communities of color, perhaps by strategies like providing PPE, sick leave, and testing to essential workers, and also to improving outcomes among Black and other minority patients once exposed. Black and other minority patients diagnosed with COVID-19 should be considered to be at higher risk than their White counterparts, and interventions or enhanced monitoring directed at this population may have the potential to improve outcomes and reduce inequities. More broadly, interventions that help reduce racism and discrimination, improve access to high quality hospitals, or improve the quality of care delivered to minority patients, should be prioritized.

In conclusion, COVID-19 case rates and mortality rates were markedly higher among NH Black and NH Other/Unknown race than among NH White residents, even after accounting for social and clinical risk, population density, and travel patterns during COVID-19.

## Supporting information

**S1 Fig. Flow diagram of patient inclusion and exclusion.** Abbreviations: MO = Missouri; MO DHSS = Missouri Department of Health and Social Services.
(TIF)

**S1 Table. Further breakdown of "Other" race category.** Numbers represent hospital claims rather than unique individuals, since patients could have one race reported on one claim and a different one reported on another claim. If a patient had an even number of claims with two different races (for example, two with White and two with Asian), he or she was left in the "other" category because we could not determine which was the appropriate category.
(DOCX)

**S2 Table. Impact of additional adjustment on ICC associated with race variable.** Abbreviations: ICC, intraclass coefficient; SDOH, social determinants of health; SE, standard error.
(DOCX)

## Acknowledgments

We thank Daniel Johnson for assistance with manuscript preparation and formatting.

## Author Contributions

**Conceptualization:** Karen E. Joynt Maddox, Mat Reidhead, Timothy McBride, Will Ross, Joseph T. Steensma, Abigail R. Barker.

**Formal analysis:** Mat Reidhead, Joshua Grotzinger, Abigail R. Barker.

**Writing – original draft:** Karen E. Joynt Maddox.

**Writing – review & editing:** Karen E. Joynt Maddox, Timothy McBride, Aaloke Mody, Elna Nagasako, Will Ross, Joseph T. Steensma, Abigail R. Barker.

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
