## [Decision Letter · Decision Letter 0]

2 Jul 2021

PONE-D-21-03586

Understanding Contributors to Racial and Ethnic Inequities in COVID-19 Incidence and Mortality Rates

PLOS ONE

Dear Dr. Joynt Maddox,

Thank you for submitting your manuscript to PLOS ONE. After careful consideration, we feel that it has merit but does not fully meet PLOS ONE’s publication criteria as it currently stands. Therefore, we invite you to submit a revised version of the manuscript that addresses the points raised during the review process.

Please address the following journal, additional editor, and reviewer comments.

We look forward to receiving your revised manuscript.

Kind regards,

Fernando A. Wilson, PhD

Academic Editor

PLOS ONE

Additional Editor Comments:

- There is inconsistent capitalization of ‘white’ within text and Fig. 3 caption.

- I suggest adding data labels to the Fig. 3 columns to be consistent with Fig. 1.

- Confirm that the PLOS ONE guideline on Acknowledgements has been followed (link below):

https://journals.plos.org/plosone/s/submission-guidelines#loc-acknowledgments

- Please verify references follow PLOS ONE formatting requirements. For example, all journal names must be abbreviated using the NCBI database.  Please refer to the following link:

https://journals.plos.org/plosone/s/submission-guidelines#loc-references

Journal Requirements:

"I have read the journal's policy and the authors of this manuscript have the following competing interests: Dr. Joynt Maddox receives research support from the National Heart, Lung, and Blood Institute (R01HL143421) and National Institute on Aging (R01AG060935, R01AG063759, and R21AG065526), and previously did contract work for the US Department of Health and Human Services. The other authors report no disclosures."

Reviewers' comments:

Reviewer's Responses to Questions

**Comments to the Author**

1. Is the manuscript technically sound, and do the data support the conclusions?

Reviewer #1: Yes

Reviewer #2: Partly

Reviewer #3: Partly

2. Has the statistical analysis been performed appropriately and rigorously? 

Reviewer #1: Yes

Reviewer #2: Yes

Reviewer #3: Yes

3. Have the authors made all data underlying the findings in their manuscript fully available?

Reviewer #1: No

Reviewer #2: Yes

Reviewer #3: No

4. Is the manuscript presented in an intelligible fashion and written in standard English?

Reviewer #1: Yes

Reviewer #2: Yes

Reviewer #3: Yes

5. Review Comments to the Author

Reviewer #1: GENERAL COMMENTS

The authors present a well written study of racial/ethnic inequities in COVID-19 incidence and mortality. The use of state-wide data is a major strength of this manuscript and the discussion of observed disparities is generally thoughtful. I provide a few specific comments below.

MAJOR COMMENTS

1. The analysis of case distribution by race/ethnicity (fig 1) does not take advantage of individual-level information on race/ethnicity and instead only presents COVID-19 case rates by ZIP-code level %Black/African American. I believe you have the data necessary to examine case rates by individual-level race/ethnicity and should present this in addition to (or in lieu of) the current analysis. This would more directly answer the first part of the question set out in the Introduction: whether the case rates are different by race/ethnicity or if among those who are diagnosed with the disease, the mortality rate differs by race/ethnicity, or both.

2. The Discussion is generally well-written and thoughtful, but primary focuses on the results for the NH Black/African American group and does not provide a discussion for the NH Other and Hispanic groups, for which elevated risks were also observed.

a. The “NH Other” group had ORs higher than the NH Black group in the fully adjusted model. The authors should provide more discussion here. Is there any other information of who comprises this “Other” group (e.g., Asian, Pacific Islanders, American Indians)? What hypotheses do they have for the observed results for this group?

b. The mortality rate among Hispanic cases was elevated but was attenuated in further adjusted models. How do these results compare with the literature? Hispanics are large diverse group, with many different countries of ancestry. Are there additional data on the Hispanic individuals in these data or in Missouri in general that could inform why different results were (or were not) observed for these analysis (e.g., Hispanic origin, immigration status, etc.)?

OTHER COMMENTS

1. Line 133: What is meant by “race/ethnicity risk factors”? Do you mean race/ethnicity as categorical variables, or factors that vary with race/ethnicity? Please specify.

2. Lines 155-158/Fig 1: Add confidence intervals for effect estimates to the figure and include in the text for effect estimates stated (i.e., 870.4 and 1,877.9).

3. Figure1: Results for mortality per 100,000 population (grey bars in Fig 1) are not mentioned in the text (neither in the Results section or in the Statistical Analysis description). Include this in the text in these two sections.

4. Lines 233-234/Fig 3: If possible with this type of analysis, add confidence intervals for effect estimates to the figure and include in the text for effect estimates stated (i.e., 1.6, 2.4, etc.).

5. Lines 332-333: The strategies listed all relate to proximal intervention strategies (i.e., closer to the individual-level—using PPE, taking sick leave, getting tested). Earlier in the Discussion you provided very clear statements about how racism and discrimination and social determinants of health may have affected COVID-19 outcomes. As such, you could include in this sentence some potential interventions targeting these more distal factors related to health.

Reviewer #2: The authors report their analysis of covid19 infection rates and outcomes using a state-wide dataset. The following are some questions and comments:

1. Figure 1 – this is an excellent figure. However, graphically, it does not clearly show that not only are there increased case number and deaths in zip codes with higher percentage of Blacks, but also the percentage of deaths among patients diagnosed with covid increases too.

2. Figure 2 – it is interesting that there is such a discrepancy between Hispanic population and the other minorities. What do the authors think accounts for this? The data presented suggest that Hispanic and Black patients had same likelihood of being insured by Medicaid, being uninsured, and having social risk factors. Based on table 1, it would seem like the differences in outcomes was attributable to prevalence of co-morbidities? In the US, both Blacks and Hispanics have higher rates of obesity and diabetes, but in the current cohort, Hispanics appear to have lower rates of obesity and uncontrolled diabetes. Does this reflect a bias in the data capture and perhaps an issue with access to care?

3. Line 206-211 – White patients had lower levels of change in travel patterns – could this also suggest less cautious behavior, or reflect differences in “essential worker” status?

4. Did the authors consider the effect of rural location or distance from a major health care center? Those factors may also impact access to care.

5. Although the authors report a 87% match rate, there were still 9000 patients who were not matched. Based on the methods, these unmatched patients did not have a hospitalization in the prior 2 years, which suggests that they were very healthy (did not require hospitalization) or did not have access to necessary medical care.

Reviewer #3: Line Comment

25 COVID-19

32 Unclear what 99.1% represents - % of total COVID cases +/- address info?

Throughout abstract Use COVID-19, not just COVID

37 COVID-19

40 Non-Hispanic other race

Throughout Manuscript Use COVID-19, not just COVID

83 Do you ever use MPID again? If not, do not need to define abbreviation

98-102 References for sources

105 (March-June, 2020)

110 Place Non-Hispanic Other at the end of the list, since it is a catch-all category

129 Inconsistent race terminology throughout paper.

154 Line glosses over important details – how many total cases, how many with complete address information? Methods cites 73,600 cases. Need a flow diagram, and more transparent text here and in methods to understand analytic denominator of study.

155 Provide total number of deaths, and overall case fatality rate.

Fig 1. Unclear what n=xx represents. Clarify legend.

Fig 2. X axis labels crowded and difficult to read

Table 1. Age needs units, e.g., year, which implies continuity

Add number of deaths row

Race terminology does not match rest of the paper, e.g., Other/Unknown – harmonize.

Why were quintiles used in table/model? No mention in Methods why/how you collapsed categories.

Do you mean to say 32.6% of people spent less time at home in 2020 vs 2019, or that 67.4% spent more time at home during COVID-19.

179 “Matched Patients” is a confusing term in the title. Remove term, flow chart would clarify your analytic sample.

199 Limit P value to < 0.001

201, 206 Starting consecutive paragraphs with “In terms of…” is awkward – change both.

Figure 3 Narrow brackets showing bars that you are comparing – appears that you are comparing blue and orange bars, but does not line up exactly

Table 2 Age needs units label, e.g., years. Why was age modeled as a continuous linear variable? Many other models find it is not linear, and use categories to define age risk, e.g., King PLOS One 2020, or splines, e.g., Mathur Lancet 2021.

251 “… white or Hispanic...”?

252 You are now being vague about a precise terminology for race category used throughout your paper.

276-291 After stating that the reasons for this are unclear, the authors charge ahead into a discussion about racism as the etiology. They never mention the term disparities a description, instead jumping to racism as a causal explanation. The underlying theme seems to be worse outcomes for Blacks = racism.

313 Add Rentsch 2021 PLOS Medicine paper for another take on VA race issue.

315 Do you mean “laboratory data” when you state “clinical data”? Rentsch paper did not use laboratory data, but rather pre-existing diagnoses, the same as your analysis.

327 Line implies that structural racism is casual in this paper, which the authors do not prove, and admit there are other explanations. Suggest disparities – a descriptor, rather than racism, a hypothesized cause.

What about patients dropped who could not be matched? 2018 census data, with population shifts in the interim? 2015 Area Depravation Index data, with changes over the last 5 years? Those without cell phones? Lack of outpatient diagnoses to determine comorbidities. How reliable is your death data?

Throughout the document, authors are inconsistent about the use COVID vs COVID-19. Change all COVID to COVID-19.

Reference 22 has odd format, prune.

6. PLOS authors have the option to publish the peer review history of their article (what does this mean?). If published, this will include your full peer review and any attached files.

Reviewer #1: No

Reviewer #2: No

Reviewer #3: No

---

## [Author Response · Author response to Decision Letter 0]

16 Sep 2021

We very much appreciate the thoughtful and thorough reviews from your reviewers and editors. In response, we have removed the ZIP-level analyses and now present only the linked patient-level analyses in the manuscript to increase clarity. We have also updated Figure 1 to reflect these changes. Lastly, we have added additional text to the discussion and limitations regarding the Non-Hispanic Other Race category and Hispanic individuals. We think these changes have meaningfully strengthened the paper and hope that you agree.

---

## [Editor Report · Decision Letter 1]

8 Nov 2021

Understanding Contributors to Racial and Ethnic Inequities in COVID-19 Incidence and Mortality Rates

PONE-D-21-03586R1

Dear Dr. Joynt Maddox,

We’re pleased to inform you that your manuscript has been judged scientifically suitable for publication and will be formally accepted for publication once it meets all outstanding technical requirements.

Kind regards,

Fernando A. Wilson, PhD

Academic Editor

PLOS ONE
---

## [Editor Report · Acceptance letter]

19 Nov 2021

PONE-D-21-03586R1 

Understanding Contributors to Racial and Ethnic Inequities in COVID-19 Incidence and Mortality Rates 

Dear Dr. Joynt Maddox:

I'm pleased to inform you that your manuscript has been deemed suitable for publication in PLOS ONE. Congratulations! Your manuscript is now with our production department. 

Kind regards, 

on behalf of

Dr. Fernando A. Wilson 

Academic Editor

PLOS ONE